# Soybean-MVS: Annotated Three-Dimensional Model Dataset of Whole Growth Period Soybeans for 3D Plant Organ Segmentation

Yongzhe Sun [1], Zhixin Zhang [1], Kai Sun [1], Shuai Li [1], Jianglin Yu [1], Linxiao Miao [1], Zhanguo Zhang [2], Yang Li [2], Hongjie Zhao [2], Zhenbang Hu [3], Dawei Xin [3], Qingshan Chen [3] and Rongsheng Zhu [2,*]

[1] College of Engineering, Northeast Agricultural University, Harbin 150030, China; 18545155636@163.com (Y.S.); s210702060@neau.edu.cn (Z.Z.); kaisunneau@163.com (K.S.); zz-3280@163.com (S.L.); jianglinyuneau@163.com (J.Y.); s220701045@neau.edu.cn (L.M.)

[2] College of Arts and Sciences, Northeast Agricultural University, Harbin 150030, China; neauzzg@neau.edu.cn (Z.Z.); code77@163.com (Y.L.); zhaohongjie77@neau.edu.cn (H.Z.)

[3] College of Agriculture, Northeast Agricultural University, Harbin 150030, China; zbhu@neau.edu.cn (Z.H.); dawxin@neau.edu.cn (D.X.); qschen@neau.edu.cn (Q.C.)

[*] Correspondence: rshzhu@126.com; Tel.: +86-133-9451-6944

**Abstract:** The study of plant phenotypes based on 3D models has become an important research direction for automatic plant phenotype acquisition. Building a labeled three-dimensional dataset of the whole growth period can help the development of 3D crop plant models in point cloud segmentation. Therefore, the demand for 3D whole plant growth period model datasets with organ-level markers is growing rapidly. In this study, five different soybean varieties were selected, and three-dimensional reconstruction was carried out for the whole growth period (13 stages) of soybean using multiple-view stereo technology (MVS). Leaves, main stems, and stems of the obtained three-dimensional model were manually labeled. Finally, two-point cloud semantic segmentation models, RandLA-Net and BAAF-Net, were used for training. In this paper, 102 soybean stereoscopic plant models were obtained. A dataset with original point clouds was constructed and the subsequent analysis confirmed that the number of plant point clouds was consistent with corresponding real plant development. At the same time, a 3D dataset named Soybean-MVS with labels for the whole soybean growth period was constructed. The test result of mAccs at 88.52% and 87.45% verified the availability of this dataset. In order to further promote the study of point cloud segmentation and phenotype acquisition of soybean plants, this paper proposed an annotated three-dimensional model dataset for the whole growth period of soybean for 3D plant organ segmentation. The release of the dataset can provide an important basis for proposing an updated, highly accurate, and efficient 3D crop model segmentation algorithm. In the future, this dataset will provide important and usable basic data support for the development of three-dimensional point cloud segmentation and phenotype automatic acquisition technology of soybeans.

**Keywords:** 3D reconstruction; the whole growth period; soybean; point cloud segmentation; dataset



## 1. Introduction

With the continuous development of plant phenomics, three-dimensional plant phenotypic analysis has become a challenging research topic. Using deep learning for point cloud segmentation is the foundation of crop phenotype measurement and breeding. The common point cloud datasets used for training are scarce and difficult to obtain, and there is no commonly used basic data for organ instance segmentation for phenotype extraction. In addition, due to the complex structure of plants, the data annotation work needs considerable manual processing. A well-labeled dataset is essential for the segmentation of plant point clouds using deep learning. In order to obtain a well-labeled dataset, it should have

the following characteristics: complete plant structure, high precision, and the ability to cover multiple varieties and growth periods. Consequently, building a labeled crop plant point cloud dataset of the entire growth period is a key step toward achieving accurate crop point cloud segmentation using deep learning.

Although the lack of well-labeled 3D plant datasets limits the further progress of plant point cloud segmentation [1], many scholars have made significant advancements in building plant point cloud segmentation datasets in recent years. Zhou et al. [2] manually segmented the 3D point cloud data of soybean plants and gave each point a real label. This was used as the training set for point cloud segmentation and real ground data for evaluating segmentation accuracy using machine learning methods. Li et al. [3] used the MVS-Pheno platform to obtain multi-view images and point clouds of corn plants in the study of organ-level point cloud automatic segmentation of corn branches based on high-throughput data acquisition and deep learning. At the same time, the research team developed a data annotation tool kit specifically for corn plants, called Label3DMatch, and annotated the data to ultimately build a training dataset. Conn et al. [4] planted tomatoes, tobacco, and sorghum under the five growth conditions of ambient light, shade, high temperature, strong light, and drought, and performed 3D laser scanning (311 tomato scans, 105 tobacco scans, and 141 sorghum scans) on the plant stem structure during 20–30 days' development. A 3D plant dataset was constructed after summarizing the species, conditions, and time points. Li et al. [5] used this original dataset and manually marked the semantic labels belonging to stems and leaves using the semantic segmentation editor (SSE) tool and established a well-labeled point cloud dataset for plant stem leaf semantic segmentation and leaf instance segmentation. Hideaki et al. [6] proposed a 3D phenotype platform that can measure plant growth and environmental information in a small indoor environment to obtain plant image datasets. In addition, annotation tools were introduced, which can manually, but effectively, create leaf tags in plant images on a pixel-by-pixel basis. Barth et al. [7] rendered a composite dataset containing 10,500 images through Blender. The scene used had 42 program-generated plant models and random plant parameters. These parameters were based on 21 empirically measured plant characteristics at 115 locations on 15 plant stems. The fruit model was obtained through 3D scanning and the plant part textures were collected through photos as a reference dataset for modeling and evaluating the segmentation performance. David et al. [8] established a large, diverse, and well-labeled wheat image dataset, called the Global Wheat Head Detection (GWHD) dataset. It contained 4700 high-resolution RGB images from multiple countries and 190,000 wheat head markers at different growth stages, with a wide range of genotypes. Wang et al. [9] constructed a lettuce point cloud dataset consisting of 620 real and synthetic point clouds fused together for 3D instance segmentation network training. Lai et al. [10] first used the SfM-MVS method to obtain point clouds of these plant population scenes, which were then annotated similarly to the S3DIS dataset to obtain data that could be trained and tested. In order to provide important and available basic data support for the development of three-dimensional point cloud segmentation and phenotype automatic acquisition technology of soybeans, this study uses the multiple-view stereo technology to construct 102 soybean three-dimensional plant models by taking advantage of its low cost, fast speed and high precision. At the same time, it is manually labeled to construct the dataset for point cloud segmentation. Compared with other datasets, this dataset contains three-dimensional information on soybean plants during the whole growth period, which has certain advantages in model accuracy and quantity.

There are several key binocular stereovision spatial positioning technologies involving image acquisition, camera calibration, image preprocessing, edge feature extraction, and stereo matching. Multi-vision is based on binocular vision, adding one or more cameras as a measuring assistant so that multiple pairs of images from different angles of the same object can be obtained. For the 3D reconstruction of a single plant, this method is more suitable for low sunlight conditions in the laboratory (Duan et al. [11]; Hui et al. [12]). This method can also be used for 3D reconstruction in the field such as studying overall crop

canopy volumes (Biskup et al. [13]; Shafiekhani et al. [14]). Compared with other methods, the multiple-view stereo method requires relatively simple equipment, and the model can be established quickly and effectively, with minimum human-computer interaction required. Although the reconstruction speed is average and the requirements for the reconstruction of environmental factors are high, the reconstruction accuracy is high, it is easy to use, and the required equipment price is relatively low. Zhu et al. [15] built a soybean digital image acquisition platform based on the principle of constructing a multi-perspective stereovision system with digital cameras covering different angles, effectively improving the problem of mutual occlusion between soybean leaves. The morphological sequence images of target plants for 3D reconstruction were then obtained. Nguyen et al. [16] described a field 3D reconstruction system for plant phenotype acquisition. The system used synchronous, multi-view, high-resolution color digital images to create real 3D crop reconstructions and successfully obtained the plant canopy geometric characteristic parameters. Lu et al. [17] developed an MCP-based SfM system using multiple-view stereo technology and studied the appropriate 3D reconstruction method and the optimal shooting angle range. Choudhury et al. [18] devised the 3DPhenoMV method. Plant images captured from multiple side views were used as the algorithm input, and a 3D model of the plant was reconstructed using multiple side views and camera parameters. Miller et al. [19] used low-cost hand-held cameras and SfM-MVS to reconstruct a spatially accurate 3D model of a single tree. Shi et al. [20] adopted the multi-view method, allowing information from two-dimensional (2D) images to be integrated into the three-dimensional (3D) plant point cloud model, and evaluated the performance of 2D and multi-view methods on tomato seedlings. Lee et al. [21] proposed an image-based 3D plant reconstruction system based on multiple UAVs to simultaneously obtain two images from different views of plants during growth and reconstruct 3D crop models with moving structures, based on multiple view stereo algorithms and metric structures. Sunvittayakul et al. [22] developed a platform for acquiring 3D cassava root crown (CRC) models using close-range photogrammetry for phenotypic analysis. This novel method is low cost, and it is easy to set up the 3D acquisition requiring only a background sheet, a reference object, and a camera and is suitable for field experiments in remote areas. Wu et al. [23] developed a small branch phenotype analysis platform, MVS-Pheno V2, based on multi-view 3D reconstruction, which focused on low plant branches and realized high-throughput 3D data collection.

In this study, the multiple view stereo method (MVS) was used to reconstruct soybean plants. A soybean image acquisition platform was constructed to obtain multi-angle images of soybean plants at different growth stages. Based on the silhouette contour principle, the model was established by contour approximation, vertex analysis, and triangulation, and 3D point cloud and original soybean datasets were constructed. Meanwhile, the obtained 3D models of soybean were manually labeled using CloudCompare v2.6.3 software. An annotated 3D dataset called Soybean-MVS, including 102 models, was established. Due to the inherent changes in the appearance and shape of natural objects, the segmentation of plant parts was a challenge. In this paper, to verify the availability of this dataset, RandLA-Net and BAAF-Net point cloud semantic segmentation networks were used to train and test the Soybean-MVS dataset.

## 2. Materials and Methods

### 2.1. Method Process

In 2018 and 2019, we cultivated high-quality soybean plants including DN251, DN252, DN253, HN48, and HN51 varieties. An original 3D soybean dataset and labeled 3D soybean plant dataset were constructed for the whole soybean growth period, consisting of the first trifoliolate stage (V1), second trifoliolate stage (V2), third trifoliolate stage (V3), fourth trifoliolate stage (V4), fifth trifoliolate stage (V5), initial flowering stage (R1), full bloom stage (R2), initial pod stage (R3), full pod stage (R4) initial seed stage (R5), full seed stage (R6), initial maturity stage (R7), and full maturity stage (R8). Among them, V represents the vegetative growth stage and R represents the reproductive growth stage. Table 1

shows the basic characteristics of experimental soybean materials, including soybean varieties, growing days, planting methods, and active accumulated temperature greater than 10 °C. The research process of this paper mainly involved 3D reconstructions based on the multiple view stereo method, manually labeling data to build datasets, and training and evaluating datasets through point cloud segmentation. Figure 1 details the overall process of building a soybean 3D dataset for point cloud segmentation.

**Table 1.** Basic characteristics of soybean materials. This shows the basic attribute information of soybean materials selected for this experiment, including soybean varieties, childbearing days, accumulated temperature and planting methods.

| Variety | Childbearing Days | >10 °C Accumulated Temperature | Planting Method |
|---|---|---|---|
| DN 251 | 125 | 2600 °C | potted planting |
| DN 252 | 124 | 2500 °C | potted planting |
| DN 253 | 115 | 2350 °C | potted planting |
| HN 48 | 118 | 2350 °C | potted planting |
| HN 51 | 126 | 2600 °C | potted planting |

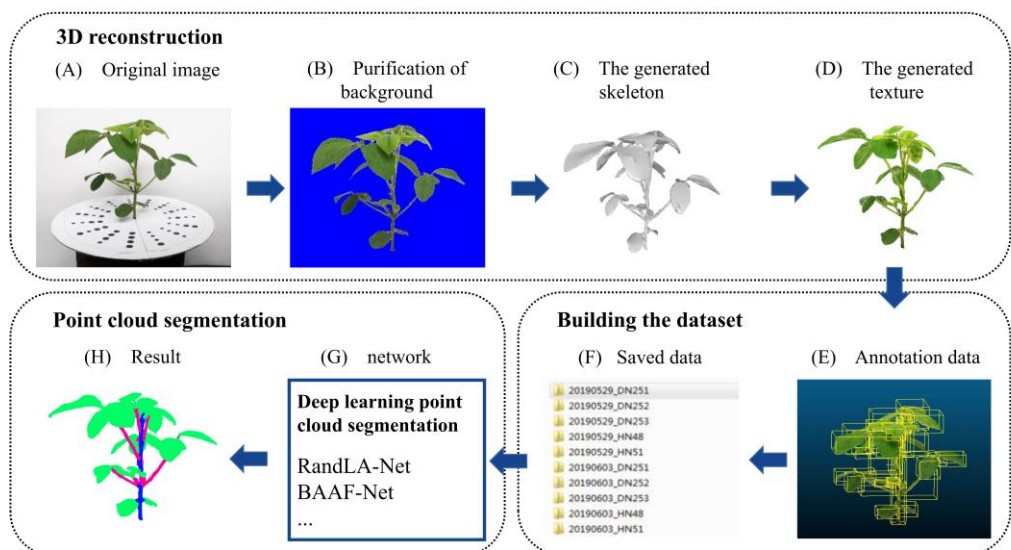

**Figure 1.** The process of building a soybean 3D dataset for point cloud segmentation. The process mainly includes three parts: 3D reconstruction, building the dataset, and point cloud segmentation. 3D reconstruction includes: (**A**) original image acquisition; (**B**) image preprocessing; (**C**) generation of 3D model skeleton; (**D**) generation of 3D model texture. Building the dataset includes: (**E**) data annotation; (**F**) construction of annotated dataset. Point cloud segmentation includes: (**G**) point cloud segmentation network selection; (**H**) result of point cloud segmentation.

*2.2. Image Acquisition*

This study prepared the image acquisition of 3D reconstruction in the room. The tools used to collect plant images included: (1) photo studio, (2) Canon EOS 600D SLR (Canon (China) Co. Ltd., Beijing, China) digital camera and camera rack, (3) rotary table, (4) calibration pad, and (5) white light absorbing cloth. A light source was added around the plant to guarantee the required basic environment needed for 3D reconstruction, based on the multiple view stereo method. The pot was about 90 cm from the camera. During the image acquisition for each pot of plants, we placed the plant pots on the rotary table, positioned a dot calibration pad at the plant roots, lowered the camera height, manually operated the rotary table, took a photo every 10°~25° (this study determined 24° according to the black dot on the calibration pad), and collected 15 photos after a circle of rotation. Then, according to the height of the plant, we adjusted the camera height three times on average, from low to high, and repeated the process. Finally, 60 photos were obtained by

taking four sets of circular rotation shots at different angles. According to the soybean growth, image acquisition was conducted at each growth stage (Figure 2). The final number of images of different varieties of soybean plants is shown in Appendix A Table A1.

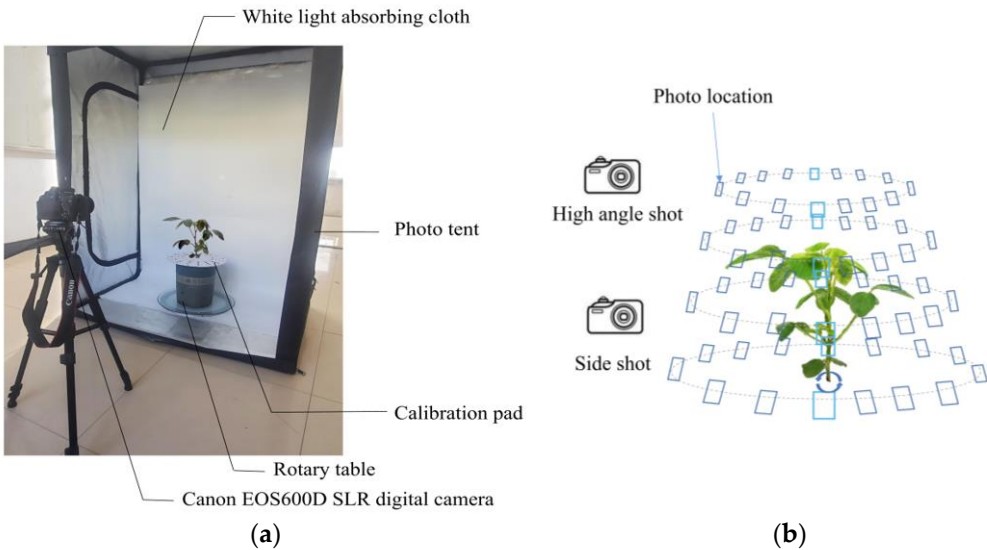

(**a**)        (**b**)

**Figure 2.** Soybean 3D reconstruction image acquisition. (**a**) Soybean image acquisition platform. (**b**) Schematic diagram of soybean plant 3D reconstruction image acquisition. The 3D reconstruction was carried out in a laboratory with no wind and sufficient light, using multiple-view stereo technology (MVS).

### 2.3. Three-Dimensional Reconstruction

This study obtained a large number of corresponding soybean plant images (about 60) from multiple perspectives. In addition, this study preprocessed basic image operations such as noise removal and distortion correction based on Python. At the same time, in the process of three-dimensional modeling, it is necessary to connect and combine images from different directions. Therefore, the relationship between the spatial positions of various images is particularly important. This study adopted the auxiliary camera calibration method of the calibration device, using a calibration pad to determine the problem of image overlap, and to determine the shooting direction of various multi-angle images. The model was established using the "contour extraction", "vertex calculation", and "visual shell generation" steps of the silhouette contour method. Silhouette contour is the contour line of the image projected on the imaging plane, which is an important clue to understanding the geometric shape of the object. When a space object is observed from multiple perspectives by perspective projection, a silhouette line of the object can be obtained in the corresponding screen of each perspective. Here the silhouette line and the corresponding perspective projection center together determine a cone of general shape in three-dimensional space, and the object to be observed is located inside this cone. By analogy, increasing the number of viewing angles of the target object from different directions can make the shape of each corresponding cone approach the surface of the object, so as to carry out three-dimensional visualization of the shape features of the target object.

Firstly, we masked the multi-angle images, selected the position of the soybean plants in each image, and purified all the background and calibration pad areas unrelated to the soybean plants, leaving only the complete soybean plant information. Then, according to the partial information of the target object in each multi-angle image, we obtained several approximate polygonal contours, numbered each approximate contour, calculated three vertices from the polygon contour, and recorded the information of each vertex. A triangular grid was used to divide the complete surface to outline the surface fine joints. The above is the realization of the "contour extraction" and "vertex calculation and visual shell generation" steps of the silhouette contour method. At that point, only the soybean

plant skeleton had been generated. In addition, further optimization operations such as volume optimization and surface refinement were required to obtain the final soybean plant surface morphology model. Finally, according to the corresponding orientation information characteristics of the three-dimensional surface contour soybean plant model obtained above, combined with the orientation information of different multi-angle images, texture mapping of its surface was performed, so that the model had more visual features and better described the characteristics of actual objects. Following three-dimensional reconstruction, 102 original models were obtained and named according to the year, date, and variety.

### 2.4. Data Annotation

The data annotation work in this study was completed using the open-source software CloudCompare v2.6.3. The acquired soybean 3D plant model (.obj format file) was imported into CloudCompare software, the leaves, main stems, and stems were manually segmented and marked on the soybean plants, and each point cloud was given a real label. At the same time, each segmented and marked organ was sampled points on a mesh. The number of sampling points was fixed at 50,000. The labeled point cloud information included xyzRGB information and was stored in .txt format. The soybean plant leaves, main stems, and stems were marked, as shown in Figure 3 (using 20180612_HN48 as an example). Finally, a labeled soybean 3D point cloud dataset named Soybean-MVS was constructed, including 102 3D models, of which 89 models were used as the training set and 13 models were used as the test set.

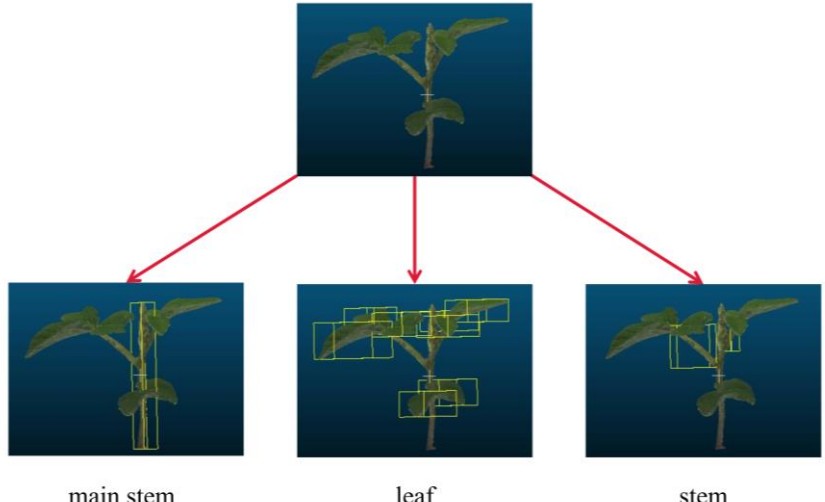

**Figure 3.** Manually mark leaves, main stems, and stems of soybean plants. The organs of the soybean plants were manually labeled.

### 2.5. Point Cloud Segmentation Network

For the semantic segmentation of the soybean-MVS 3D point cloud dataset, this study selected two deep learning-based point cloud segmentation network architectures, (1) RandLA-Net [24]; (2) BAAF-Net [25] to test its availability. Appendix A Table A2 shows the hardware, software, and super parameter configuration of the deep learning model. Figure 4 shows the architecture of the two-point cloud segmentation semantic models. We have already submitted the data and computer programs used for the analysis, which will allow the results of our experiments to be reproduced by anyone. The link addresses are https://github.com/18545155636/BAAF-Net.git (accessed on 1 January 2023) and https://github.com/18545155636/randla-net.git (accessed on 1 January 2023). The following briefly describes the key methods of these architectures for encoding 3D point cloud local geometry. Please refer to the original text for the default structure and other details of the architecture.

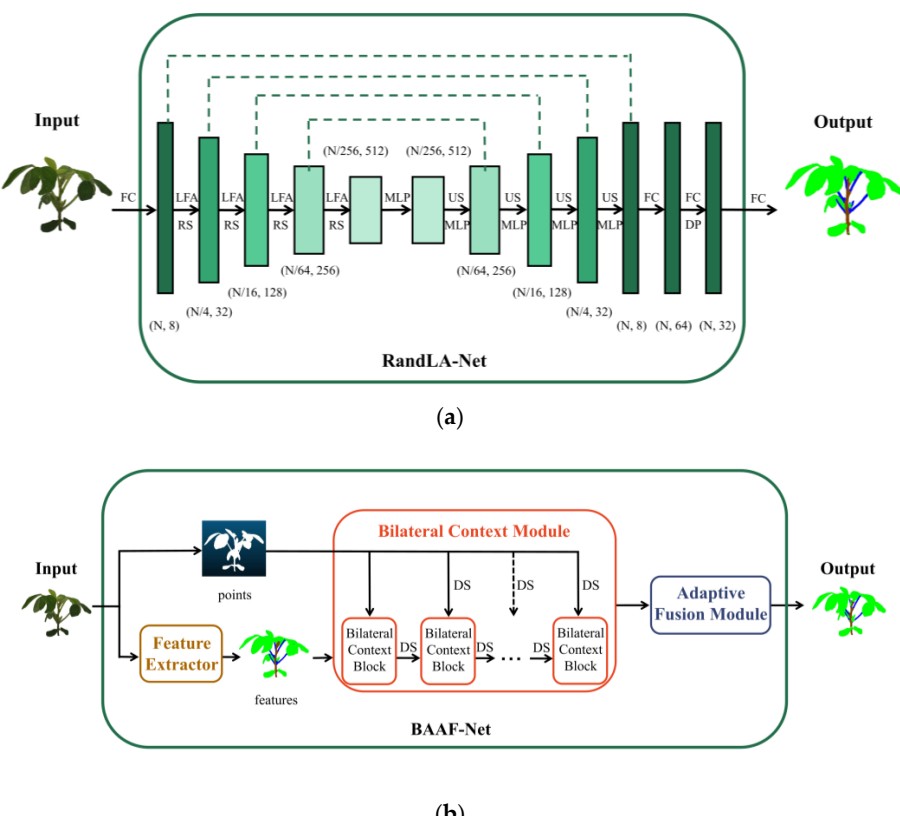

(**a**)

(**b**)

**Figure 4.** Point cloud semantic segmentation architecture. (**a**) RandLA-Net semantic segmentation architecture diagram. (**b**) BAAF-Net semantic segmentation architecture diagram. The dataset was trained and tested on two networks.

### 2.5.1. RandLA-Net

RandLA-Net is an effective and portable network that can identify the semantics of each point and apply it to large-scale point clouds. It uses the local feature aggregation module (LFA) to gradually improve the receptivity of each 3D point, which can effectively save the geometric details of the point cloud. The local feature aggregation module involves three main steps:

The first step is local spatial encoding (LocSE). The coordinates and features of a point (center point) in the point cloud P and K points adjacent to the point are taken as input. It consists of three parts: (1) Finding neighboring points, (2) relative point position encoding, and (3) point feature augmentation. A new adjacent feature of the center point is output, which encodes the local geometric feature of the center point. This module can significantly learn the local geometric features of point clouds, which will eventually play a beneficial role in learning the complex local structure information of the entire network. The second step is known as attention pooling. The LocSE output is used as the input of this step. This includes two parts: (1) computing attention scores and (2) weighted summation. Then, the feature vectors generated by the center point aggregated local features are output. The third step is called the divided residential block. It consists of multiple LocSE and attention pooling layers plus a skip connection.

RandLA-Net regards each point as the center point and each point aggregates the information of the surrounding points to itself. According to the principle that the points sampled in the whole point cloud by random sampling should conform to a normal distribution, random sampling is directly adopted. By employing this, the sampling speed can be greatly accelerated.

2.5.2. BAAF-Net

BAAF-Net uses a bilateral structure to increase the local context information of a point, while adaptively fusing multi-resolution features, to propose a new point cloud semantic segmentation network, involving the following two steps:

The first step is the bilateral context module. This consists of multiple bilateral context blocks (BCBs). A BCB is composed of bilateral augmentation and mixed local aggregation. During bilateral augmentation, the neighborhood information is aggregated around a point to the point to obtain the local context information in the geometric and feature spaces, but this is insufficient to express the domain information. Then, the local geometric context information is adjusted through the local semantic context information, which in turn is adjusted through the enhanced local geometric context information. Finally, MLP is used to further process the enhanced local geometric, and local semantic, context information and stack them together to obtain the enhanced local context information. The mixed local aggregation process uses the maximum pooling method, that is, the maximum K values of each feature are calculated as the value of the feature of point i. Then, the mean point of the local neighborhood of the point is learned through MLP, and the feature of the point is taken as the feature of point i. Lastly, the above two aggregated features are spliced to obtain the final feature of point i. The bilateral context module is used to combine bilateral context modules and continuously output the downsampled points to BCB, which is also the corresponding encoder part. The second step is the Adaptive Fusion Module. This part corresponds to the decoder. The encoder will output feature maps with different resolutions. The output of each layer is gradually upsampled to obtain full-size feature maps. The previous layer's feature maps need to be fused each time upsampling is performed. Then, the full-size feature maps sampled on these multiple scales need to be fused. To obtain different-sized important information, the full-size feature map is inputted into MLP to obtain the point level information, which is then normalized using Sofmax. Finally, the integrated feature map for semantic segmentation is obtained by fusing the normalized point level information and the full-size feature map after upsampling.

BAAF-Net enhances its local context by making full use of geometric and semantic features in bilateral structures. It fully explains the uniqueness of points from multiple resolutions and represents feature maps at the point level according to adaptive fusion methods for accurate semantic segmentation.

*2.6. Evaluation Index*

In this study, the average value of the *IoU* scores of three categories (*mIoU*) and the average accuracy (*mAcc*) were used to evaluate the success of each architecture. The number of true positives, true negatives, false positives, and false negatives in each category were expressed as *TP*, *TN*, *FP*, and *FN*, respectively. Then, the intersection over union (*IoU*) of each semantic class, the total accuracy (*Acc*) of each plant, the mean score of *IoU* (*mIoU*), and the mean accuracy (*mAcc*) were defined as:

$$Accuracy = \frac{TP + TN}{TP + TN + FP + FN}, \tag{1}$$

$$IoU = \frac{TP}{TP + FP + FN}, \tag{2}$$

$$mAcc = \frac{1}{n}\sum_{i=1}^{n} Acc, \tag{3}$$

$$mIoU = \frac{1}{k}\sum_{i=1}^{k} IoU, \tag{4}$$

where *n* represents the total number of datasets in the test set (13 data) and *k* represents the total number of categories.

## 3. Results

### *3.1. Soybean-MVS Dataset*

#### 3.1.1. Original 3D Dataset

This paper tracked and recorded the entire growth period of five varieties of soybean and created a 3D reconstruction of the soybean plants during each period. A total of 102 3D virtual soybean plants were obtained and a 3D point cloud dataset of original soybean plants was constructed. Appendix A Table A3 details the point cloud of the original soybean 3D plant dataset. Figure 5 shows the point cloud information map of the original soybean three-dimensional plant dataset. Figure 5a displays the comparison results of the total point cloud cover of stage V and stage R using a *t*-test. It can be seen that there was a significant difference between the point cloud covers of stage R and stage V, with the stage R point cloud cover being significantly larger than that of stage V. Figure 5b shows the comparison results of the reconstructed point cloud cover in 2018 and 2019 using a *t*-test. It can be seen that the reconstructed model had almost the same point cloud cover over two years. Figure 5c is the comparison map of the point cloud cover of soybean plants at different development stages following an ANOVA variance test, among which the point cloud cover of soybean plants at the R5 stage is the greatest, indicating that soybean plants grow the most vigorously during the R5 stage and reach the peak stage of their development. The two control graphs show that the more complex the soybean plant, the greater the model point cloud cover. Figure 5d is the comparison map of point cloud cover of different soybean varieties after an ANOVA variance test, and the difference in point cloud cover among different varieties is not found to be significant.

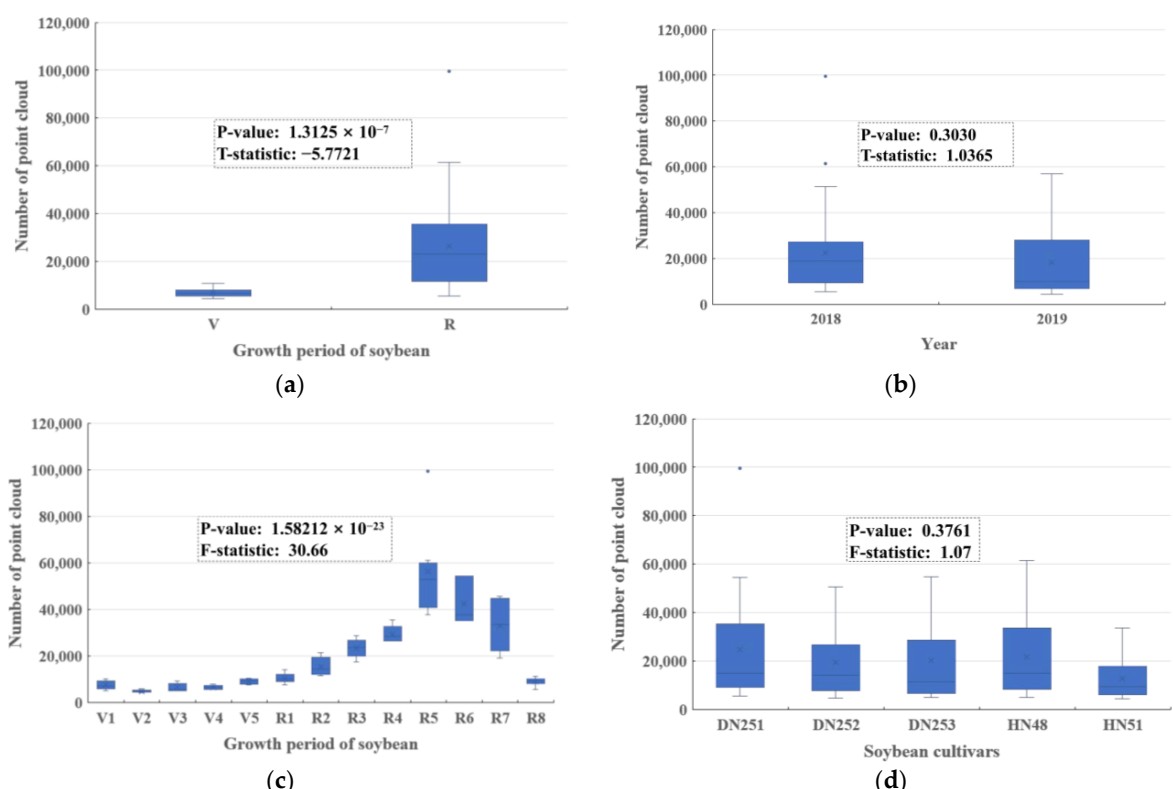

**Figure 5.** Point cloud information map of original soybean 3D plant dataset. (**a**) Comparison chart of total point cloud amount of stage V and stage R. (**b**) Comparison chart of reconstructed point cloud amount in 2018 and 2019. (**c**) Comparison chart of point cloud amount in different development stages. (**d**) Comparison chart of point cloud amounts of various varieties.

### 3.1.2. Labeled 3D Dataset

This study annotated the original dataset. In order to homogenize the point cloud, this study conducted network point collection for each labeled organ, and the number of sampled point clouds was controlled at 50,000. A labeled soybean 3D point cloud dataset was constructed. Figure 6 compares the point amount of the original 3D dataset and the sampled point cloud amount of the labeled 3D dataset, taking the DN252 soybean plant as an example. The leaves, main stems, and stems of three soybean plant organs were manually marked. Table 2 shows the number of organs of different types of soybean plants after labeling.

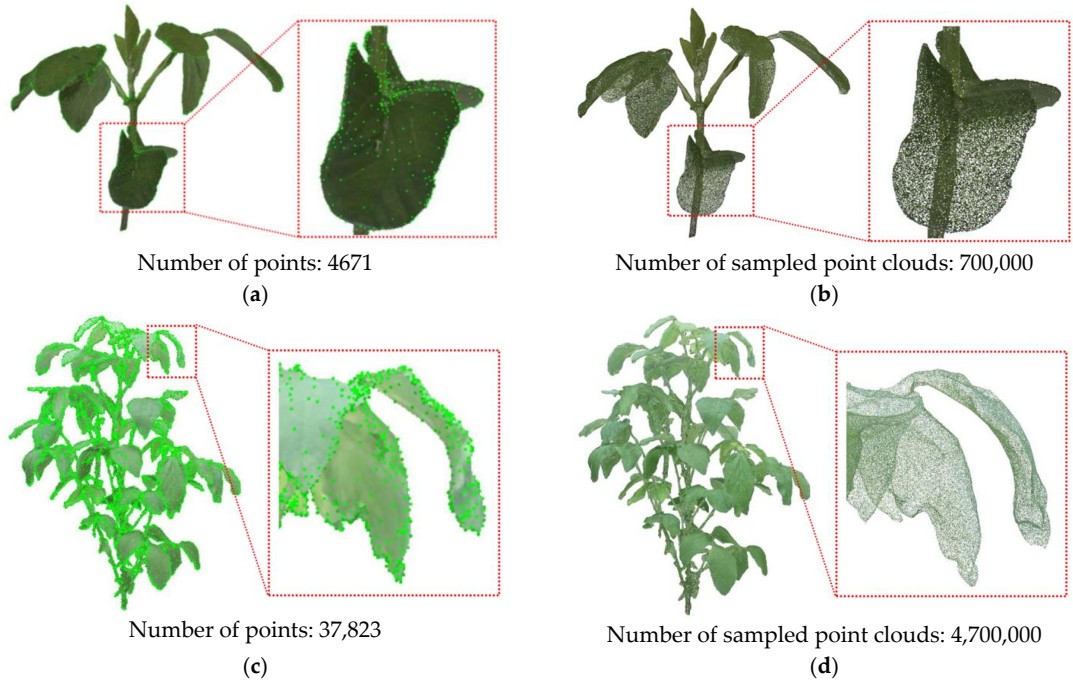

Number of points: 4671
(**a**)

Number of sampled point clouds: 700,000
(**b**)

Number of points: 37,823
(**c**)

Number of sampled point clouds: 4,700,000
(**d**)

**Figure 6.** Comparison between the amount of points in the original 3D dataset and the amount of sampled point clouds in the labeled 3D dataset. (**a**,**c**) Point volume of the original dataset. (**b**,**d**) Sampled point cloud volume of labeled dataset.

**Table 2.** Number of organ markers in different soybean plants. The number of leaves, the number of main stems, and the number of stems were compared by counting the organs of labeled soybean plants.

|  | Leaf | Main Stem | Stem |
| --- | --- | --- | --- |
| DN251 | 756 | 22 | 182 |
| DN252 | 813 | 22 | 188 |
| DN253 | 718 | 20 | 165 |
| HN48 | 649 | 21 | 161 |
| HN51 | 437 | 17 | 125 |

Finally, 89 labeled models were divided into a training set, and 13 labeled models were divided into a test set. The point cloud amount distribution of each organ in the training set and test set is shown in Table 3.

**Table 3.** Point cloud amount distribution of each organ in the training set and test set (%). The proportion of cloud cover of different organ points in the training set and the test set was calculated.

|  | Leaf | Main Stem | Stem |
| --- | --- | --- | --- |
| Soybean-MVS training models | 78.08 | 2.72 | 19.20 |
| Soybean-MVS test models | 79.13 | 2.36 | 18.51 |

### 3.2. Point Cloud Segmentation

The test results of 20 models in the Soybean-MVS dataset on the RandLA-Net and BAAF-Net models are shown in Table 4.

**Table 4.** Point cloud segmentation test results (%). The results of the dataset on two models, including IoU, mIoU, and mAcc.

|  |  | **RandLA-Net** | **BAAF-Net** |
|---|---|---|---|
|  | leaf | 88.58 | 88.83 |
| IoU | main stem | 57.03 | 27.25 |
|  | stem | 45.54 | 48.23 |
| mIoU |  | 63.72 | 54.77 |
| mAcc |  | 88.52 | 87.45 |

Figure 7 shows the Acc of the same soybean plant (DN251) at different growth stages after RandLA-Net and BAAF-Net network tests. Overall, the mAcc tested by the two networks was high. For the different complex stages of soybean plant growth, the segmentation accuracy was high and there was no significant difference. Among them, the Acc value in the R5 period was the highest, which may be because the soybean plants are the most vigorous and the leaves are the most luxuriant during the R5 period. The effect of the two networks on the leaf segmentation was better than on the main stems and stems. At the R8 stage, because the soybean plant was leafless, the Acc value was lowest.

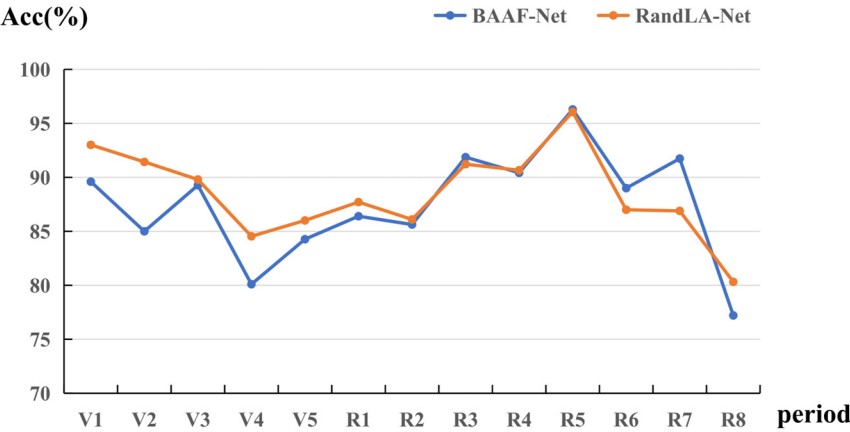

**Figure 7.** Acc results of soybean plants tested in the RandLA-Net network and BAAF-Net network during the whole growth period. This shows a comparison of the Acc results of the test set on the two models.

Figure 8 shows the label data, label data visualization results, RandLA-Net test visualization results, and BAAF-Net test visualization results of the DN251 soybean plants. From the results, both networks separated soybean plant leaves, main stems, and stems, but there were still identification errors in some details. Figure 9 highlights an example of a false prediction with a red ellipse. In terms of leaves, both networks performed well, which may be due to the regular leaf shape and a large amount of training, and they were all segmented. However, Figure 9a,b show that the two networks recognized stems as leaves when recognizing the petiole. In terms of the main stem, BAAF-Net performed worse than RandLA-Net. Figure 9c,d show that some main stem components were identified as stems. This may be due to the small amount of main stem training and the similar morphology of main stems and stems. In terms of the stem, Figure 9e,f show that both network test results identified the stems as part as leaves. In addition, Figure 9g,h show that RandLA-Net identified the connection between main stems and stems as a leaf, while the BAAF-Net performed well.

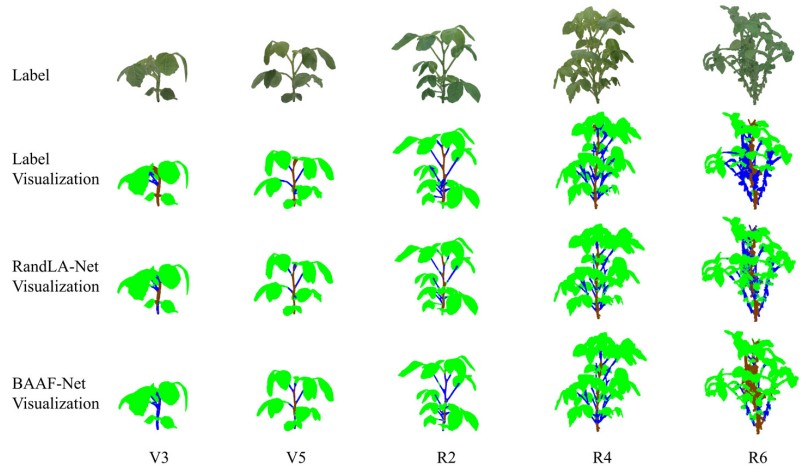

**Figure 8.** Soybean plant annotation data, RandLA Net, and BAAF Net visualization results in different stages. By contrast, this shows the overall segmentation effect of the two models.

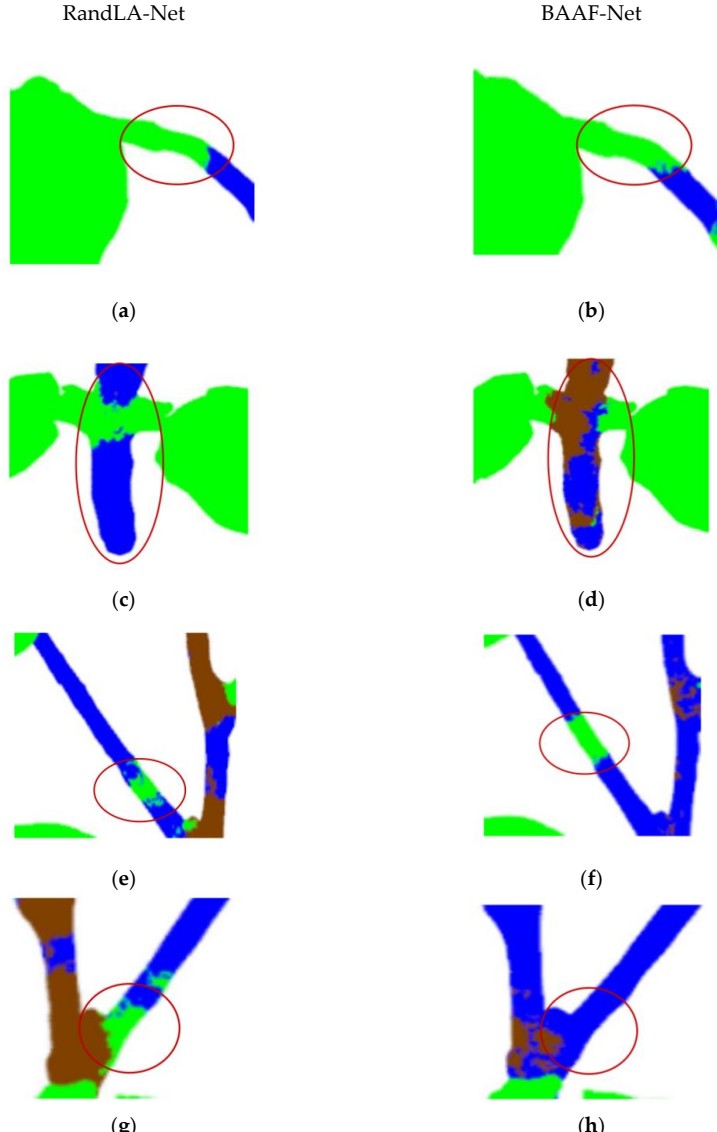

**Figure 9.** Example of error prediction. (**a,b**) Examples of false prediction of the petiole. (**c,d**) Examples of main stem error prediction. (**e,f**) Examples of stem error prediction. (**g,h**) Examples of error prediction at

the connection of main stem and stem. (**a**,**c**,**e**,**g**) The RandLA-Net test results. (**b**,**d**,**f**,**h**) BAAF-Net test results. By contrast, this shows the local segmentation difference between the two models.

## 4. Discussion

This paper explored the growth of soybean plants based on 3D reconstruction technology. Figure 10 shows the full soybean plant growth period, using the three-dimensional model of DN251 soybean plants constructed in this study as an example. The original three-dimensional soybean plant whole growth period dataset and the labeled three-dimensional plant soybean whole growth period dataset constructed in this study can provide an important basis for solving and tackling issues raised by breeders, producers, and consumers. For example, research on crop phenotypic measurement and other issues requires the effective phenotypic analysis of plant growth and morphological changes throughout the growth period. Considering this, we propose the use of point cloud segmentation.

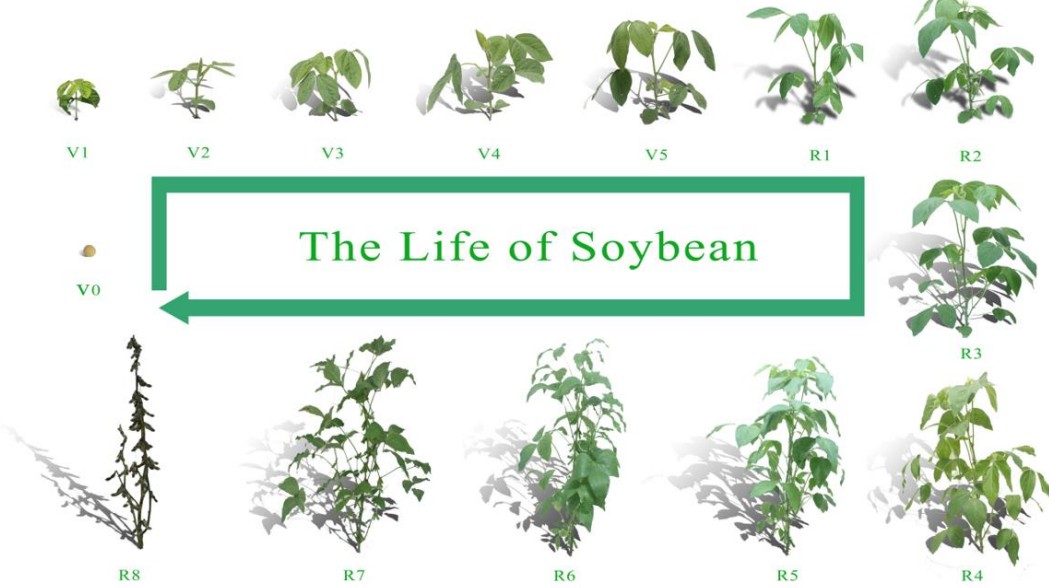

**Figure 10.** The life of soybean.

First of all, this paper chose the multiple-view stereo method to reconstruct the entire growth period of soybean plants. This method obtains detailed information about plants through crop images and extracts the phenotypic parameters of crops through related algorithms. Cao et al. [26] developed a 3D imaging acquisition system to collect plant images from different angles to reconstruct 3D plant models. However, only 20 images were collected in that study to meet the minimum image overlap requirements for 3D model reconstruction. In our study, 60 soybean plant images from different perspectives were collected at four different heights during image acquisition, so the 3D model obtained after 3D reconstruction was more accurate. At the same time, a three-dimensional dataset of the whole growth period of the original soybean was established. By comparing the original point cloud amount of the V and R stages, the relationship between the point cloud amount of the three-dimensional soybean plant model and the growth period was analyzed, which confirmed that the number of plant point clouds was consistent with corresponding real plant development. This provides an important basis for more accurate three-dimensional reconstruction of crops in the whole growth period in the future.

Secondly, training point cloud segmentation models usually require a large amount of tag data, the cost of which is very high, particularly in intensive prediction tasks such as semantic segmentation. In addition, the plant phenotype dataset also faces the additional challenges of severe occlusion and different lighting conditions, which makes obtaining annotations more time-consuming (Rawat et al. [27]). Gong et al. [28] used a structured light

3D scanning platform, based on a special turntable, to obtain the 3D point cloud data of rice panicles, and then used the open-source software LabelMe to mark point by point and create a rice panicle point cloud dataset. Boogaard et al. [29] manually marked cucumber plants twice with CloudCompare and constructed annotated dataset A and annotated dataset B. Dutagaci et al. [30] obtained 11 3D point cloud models of Rosa through X-ray tomography and manually annotated them, creating a labeled dataset to evaluate 3D plant organ segmentation methods, called the ROSE-X dataset. However, these datasets do not emphasize the importance of three-dimensional data of the entire growth period of plants and the amount of data is relatively small, which lacks integrity for subsequent studies such as the phenotypic measurement of whole plant growth periods. In our study, Soybean-MVS, a labeled three-dimensional dataset of the whole growth period of soybean, was constructed, which fully meets the data volume requirements of in-depth learning point cloud segmentation training and evaluation and ensures the integrity of the dataset used for the point cloud segmentation research. This not only provides a basis for measuring plant phenotype, bionic species, and other issues, but may also provide a basis for exploring the natural laws of plant growth.

Thirdly, in the process of labeling the dataset in our paper, since the soybean plant main stem and stem information are relatively similar, and a soybean plant only has one main stem, the number is much lower than leaf and stem, leading to a low segmentation accuracy of the main stems. There is a situation where the points on the petiole were classified as leaves. However, the visualization results show that each point cloud segmentation network model still segmented most of the points on the main stems. Therefore, the Soybean-MVS dataset can ensure the effectiveness of the point cloud segmentation task.

Finally, the Soybean-MVS dataset is universal. The universality of datasets is crucial to empirical research evaluation for at least three reasons: (1) providing a basis for measuring progress by copying and comparing results; (2) revealing the shortcomings of the latest technology, thus paving the way for novel methods and research directions; (3) the method can be developed without first collecting and tagging data (Schunck et al. [31]). Furthermore, data with high universality can meet the requirements of different point cloud segmentation models and obtain a highly reliable segmentation model. Turgut et al. [32] evaluated their performance on real rose shrubs based on the ROSE-X and synthetic model datasets and adjusted six-point cloud-based deep learning architectures (PointNet, etc.) to subdivide the structure of a rosebush model. In our paper, RandLA-Net and BAAF-Net were used for testing (also applicable to other 3D point cloud classification and segmentation models based on depth learning). In the future, we will continue to expand and adjust the Soybean-MVS dataset and apply it to other point cloud segmentation network models, to further improve it.

## 5. Conclusions

In order to provide important and usable basic data support for the development of three-dimensional point cloud segmentation and phenotype automatic acquisition technology of soybeans, this paper adopted the multiple-view stereo technology and obtained 60 photos in each group through four different height circular rotation shots. Three-dimensional plant reconstruction was carried out using the profile contour method to construct the original three-dimensional soybean plant dataset of the whole growth period. It was concluded that the number of point clouds was consistent with the actual plant development. The leaf, mainstem and stem in the obtained data and sample points were manually annotated on a mesh. A soybean three-dimensional plant dataset named Soybean-MVS was constructed for point cloud semantic segmentation. Finally, RandLA-Net and BAAF-Net models were used to evaluate the dataset, and the mAcc of the test results were 88.52% and 87.45%, respectively. The usability of the Soybean-MVS labeled 3D plant dataset was verified. The publication of this dataset provides an important basis for proposing an updated, high-precision, and efficient 3D crop model segmentation algorithm. In the future, we will constantly update and supplement the dataset, and apply it to more point cloud

segmentation models to make it more universal. At the same time, the automatic acquisition and breeding of soybean phenotype will be further explored on the basis of this dataset.

**Author Contributions:** Y.S.: formal analysis, investigation, methodology, image acquisition, three-dimensional reconstruction, annotation of data and writing—original draft. Z.Z. (Zhixin Zhang): supervision and validation. K.S. and J.Y.: image acquisition and three-dimensional reconstruction. S.L. and L.M.: annotation of data. Y.L., Z.H., Z.Z. (Zhanguo Zhang) and H.Z.: project administration and resources. D.X.: writing—review and editing and funding acquisition. Q.C.: writing—review and editing, funding acquisition, and resources. R.Z.: designed the research of the article, conceptualization, data curation, funding acquisition, resources, and writing—review and editing. All authors agreed to be accountable for all aspects of their work to ensure that the questions related to the accuracy or integrity of any part is appropriately investigated and resolved. All authors have read and agreed to the published version of the manuscript.

**Funding:** This work was supported by the Natural Science Foundation of Heilongjiang Province of China (LH2021C021).

**Institutional Review Board Statement:** Not applicable.

**Data Availability Statement:** Original models are available in a publicly accessible repository: The original contributions presented in the study are publicly available. These data can be found here: https://www.kaggle.com/datasets/soberguo/soybean-original-model (accessed on 1 January 2023). The soybean-MVS dataset is available in a publicly accessible repository: Publicly available datasets were analyzed in this study. These data can be found here: https://www.kaggle.com/datasets/soberguo/soybeanmvs (accessed on 1 January 2023).

**Conflicts of Interest:** The authors declare no conflict of interest.

## Appendix A

**Table A1.** Image collection quantity of soybean plants of different varieties in different stages.

| | V1 | | V2 | | V3 | | V4 | | V5 | | R1 | | R2 | | R3 | | R4 | | R5 | | R6 | | R7 | | R8 | |
|---|---|---|---|---|---|---|---|---|---|---|---|---|---|---|---|---|---|---|---|---|---|---|---|---|---|---|
| | 2018 | 2019 | 2018 | 2019 | 2018 | 2019 | 2018 | 2019 | 2018 | 2019 | 2018 | 2019 | 2018 | 2019 | 2018 | 2019 | 2018 | 2019 | 2018 | 2019 | 2018 | 2019 | 2018 | 2019 | 2018 | 2019 |
| DN251 | 0 | 60 | 0 | 60 | 60 | 60 | 60 | 60 | 0 | 60 | 60 | 60 | 60 | 60 | 60 | 60 | 0 | 60 | 60 | 60 | 60 | 60 | 60 | 60 | 60 | 60 |
| DN252 | 0 | 60 | 0 | 60 | 60 | 60 | 60 | 60 | 0 | 60 | 60 | 60 | 60 | 60 | 60 | 60 | 0 | 60 | 60 | 60 | 60 | 60 | 60 | 60 | 60 | 60 |
| DN253 | 0 | 60 | 0 | 60 | 60 | 60 | 60 | 60 | 0 | 60 | 60 | 60 | 60 | 60 | 60 | 60 | 0 | 60 | 60 | 60 | 60 | 60 | 60 | 60 | 60 | 60 |
| HN48 | 0 | 60 | 0 | 60 | 60 | 60 | 60 | 60 | 0 | 60 | 60 | 60 | 60 | 60 | 60 | 60 | 0 | 60 | 60 | 60 | 60 | 60 | 60 | 60 | 60 | 60 |
| HN51 | 0 | 60 | 0 | 60 | 60 | 60 | 60 | 60 | 0 | 60 | 60 | 60 | 60 | 60 | 60 | 60 | 0 | 60 | 60 | 60 | 60 | 60 | 60 | 60 | 60 | 60 |

Notes: In this study, five kinds of soybeans, DN251, DN252, DN253, HN48 and HN51, were planted in the pot farm of Northeast Agricultural University in 2018 and 2019, and images were collected during the whole growth period of soybeans. Table 1 shows the specific number of images collected.

**Table A2.** Hardware, software, and hyperparameter configuration of deep learning models.

| Catalogue | Content |
|---|---|
| CPU | Core i9-12900kf |
| RAM | 64 GB |
| GPU | NVIDIA 3090 (24 GB) |
| operating system | Ubuntu 18.04 |
| Cuda | 11.3 |
| Cudnn | 8.4 |
| Data Annotation | CloudCompare |
| Deep learning framework | Tensorflow 2.6.0 |
| Anaconda | Anaconda 5.2 |
| Momentum | 0.9 |
| threshold | 0.5 |

**Table A3.** Original information of 3D soybean plant model.

| Variety | Date of Reconstruction | Stage | Points |
|---------|----------------------|-------|--------|
| DN251 | 12 June 2018 | V3 | 66,528 |
| DN252 | 12 June 2018 | V3 | 85,871 |
| DN253 | 12 June 2018 | V3 | 5164 |
| HN48 | 12 June 2018 | V3 | 63,915 |
| HN51 | 12 June 2018 | V3 | 5390 |
| DN251 | 19 June 2018 | V4 | 78,211 |
| DN252 | 19 June 2018 | V4 | 7482 |
| DN253 | 19 June 2018 | V4 | 6581 |
| HN48 | 19 June 2018 | V4 | 5776 |
| HN51 | 19 June 2018 | V4 | 6734 |
| DN251 | 26 June 2018 | R1 | 10,752 |
| DN252 | 26 June 2018 | R1 | 140,986 |
| DN253 | 26 June 2018 | R1 | 11,535 |
| HN48 | 26 June 2018 | R1 | 9371 |
| DN251 | 4 July 2018 | R2 | 14,842 |
| DN252 | 4 July 2018 | R2 | 21,367 |
| HN48 | 4 July 2018 | R2 | 18,757 |
| HN51 | 4 July 2018 | R2 | 12,300 |
| DN251 | 11 July 2018 | R3 | 25,306 |
| DN252 | 11 July 2018 | R3 | 24,316 |
| DN253 | 11 July 2018 | R3 | 26,733 |
| HN48 | 11 July 2018 | R3 | 22,995 |
| HN51 | 11 July 2018 | R3 | 271,221 |
| DN251 | 26 July 2018 | R5 | 99,451 |
| DN252 | 26 July 2018 | R5 | 37,704 |
| DN253 | 26 July 2018 | R5 | 51,456 |
| HN48 | 26 July 2018 | R5 | 61,301 |
| HN51 | 26 July 2018 | R5 | 808,638 |
| DN251 | 17 August 2018 | R6 | 35,193 |
| DN252 | 17 August 2018 | R6 | 37,896 |
| DN251 | 8 September 2018 | R7 | 24,864 |
| DN252 | 8 September 2018 | R7 | 19,805 |
| DN253 | 8 September 2018 | R7 | 19,145 |
| HN48 | 8 September 2018 | R7 | 35,983 |
| HN51 | 8 September 2018 | R7 | 33,647 |
| DN251 | 3 October 2018 | R8 | 5574 |
| DN252 | 3 October 2018 | R8 | 8662 |
| DN253 | 3 October 2018 | R8 | 11,313 |
| HN48 | 3 October 2018 | R8 | 11,220 |
| HN51 | 3 October 2018 | R8 | 9366 |
| DN251 | 29 May 2019 | V1 | 9415 |
| DN252 | 29 May 2019 | V1 | 10,233 |
| DN253 | 29 May 2019 | V1 | 7014 |
| HN48 | 29 May 2019 | V1 | 8766 |
| HN51 | 29 May 2019 | V1 | 6541 |
| DN251 | 3 June 2019 | V2 | 6113 |
| DN252 | 3 June 2019 | V2 | 4671 |
| DN253 | 3 June 2019 | V2 | 4860 |
| HN48 | 3 June 2019 | V2 | 4947 |
| HN51 | 3 June 2019 | V2 | 4269 |
| DN251 | 8 June 2019 | V3 | 8322 |
| DN252 | 8 June 2019 | V3 | 5228 |
| DN253 | 8 June 2019 | V3 | 5161 |
| HN48 | 8 June 2019 | V3 | 7974 |
| HN51 | 8 June 2019 | V3 | 5777 |
| DN251 | 12 June 2019 | V4 | 7890 |
| DN252 | 12 June 2019 | V4 | 5612 |
| DN253 | 12 June 2019 | V4 | 88,756 |

**Table A3.** *Cont.*

| Variety | Date of Reconstruction | Stage | Points |
|---------|------------------------|-------|--------|
| HN48 | 12 June 2019 | V4 | 113,444 |
| HN51 | 12 June 2019 | V4 | 5956 |
| DN251 | 18 June 2019 | V5 | 9132 |
| DN252 | 18 June 2019 | V5 | 7669 |
| DN253 | 18 June 2019 | V5 | 9416 |
| HN48 | 18 June 2019 | V5 | 10,604 |
| HN51 | 18 June 2019 | V5 | 100,902 |
| DN251 | 24 June 2019 | R1 | 149,372 |
| DN252 | 24 June 2019 | R1 | 9728 |
| DN253 | 24 June 2019 | R1 | 135,007 |
| HN48 | 24 June 2019 | R1 | 160,789 |
| HN51 | 24 June 2019 | R1 | 7672 |
| DN251 | 27 June 2019 | R2 | 13,951 |
| DN252 | 27 June 2019 | R2 | 171,706 |
| DN253 | 27 June 2019 | R2 | 176,975 |
| HN48 | 27 June 2019 | R2 | 242,936 |
| HN51 | 27 June 2019 | R2 | 11,597 |
| DN251 | 5 July 2019 | R3 | 19,569 |
| DN252 | 5 July 2019 | R3 | 20,336 |
| DN253 | 5 July 2019 | R3 | 286,872 |
| HN48 | 5 July 2019 | R3 | 22,544 |
| HN51 | 5 July 2019 | R3 | 17,661 |
| DN251 | 13 July 2019 | R4 | 29,729 |
| DN252 | 13 July 2019 | R4 | 26,609 |
| DN253 | 13 July 2019 | R4 | 28,611 |
| HN48 | 13 July 2019 | R4 | 35,583 |
| HN51 | 13 July 2019 | R4 | 26,426 |
| DN251 | 22 July 2019 | R5 | 37,823 |
| DN252 | 22 July 2019 | R5 | 50,636 |
| DN253 | 22 July 2019 | R5 | 54,806 |
| HN48 | 22 July 2019 | R5 | 56,830 |
| DN251 | 6 August 2019 | R6 | 54,325 |
| DN252 | 6 August 2019 | R6 | 712,682 |
| DN253 | 6 August 2019 | R6 | 632,552 |
| HN48 | 6 August 2019 | R6 | 603,497 |
| DN251 | 26 August 2019 | R7 | 45,556 |
| DN252 | 26 August 2019 | R7 | 45,332 |
| DN253 | 26 August 2019 | R7 | 44,100 |
| HN48 | 26 August 2019 | R7 | 27,986 |
| DN251 | 21 September 2019 | R8 | 9990 |
| DN252 | 21 September 2019 | R8 | 8426 |
| DN253 | 21 September 2019 | R8 | 9317 |
| HN48 | 21 September 2019 | R8 | 7229 |
| HN51 | 21 September 2019 | R8 | 9964 |

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
