# Peer review of "Soybean-MVS: Annotated Three-Dimensional Model Dataset of Whole Growth Period Soybeans for 3D Plant Organ Segmentation"

_agriculture, doi:10.3390/agriculture13071321_

Round 1
Reviewer 1 Report
This research article focused on three dimensional annotated datasets for Soybean for 3D segmentation of plant organs. The article is well flown. To improve the quality of the paper, the authors need to consider the following items for the revision.
Major comments:
1. It is suggested that the author revises and check English writing. Besides author check the author guidelines for the manuscript formatting.
2. In abstract, it is advised to add some major findings from the study and some key drawbacks/scope remain for further study. The objective of this study is absent also, revise the abstract.
3. The introduction is well flown. It is suggested to review the literature to add some more specific information. Many authors provide 3D point cloud methods and datasets. What makes the difference from them? Author need to clarify why we need this study although there are already several methods with various advantages. It is not clear in the introduction.
4. What are the real motivation of this study? Lines-77-78, is not a match in that paragraph. It is advised to revise.
5. What is table 1 for? Is it useful information or is it connected with the dataset information? Please elaborate. What is >10 ℃ accumulated temperature?
6. In image acquisition method, how the distance between camera and plant were determined? Moreover, the rotary table was manually operated, so how the degree between the images were determined? Why 24o? Most importantly, in 3D reconstruction, overlap (%) is required between images, how author maintain the image overlap (%)(both H and W) and how much it was? Image overlap (%) also an important issue for 3d point cloud generation which influence the final 3D outcomes. How author control this condition? Please elaborate.
7. It would be better to show every steps as figure as explain in lines 171-189, three dimensional reconstruction.
8. Line 191, 102 original datasets were obtained and named according to the year, date, and variety. It is not clear what kind of dataset?
9. Is equation 3 and 4 are same or different? Please revise?
10. Line 294, 5A displays the comparison results, what is 5A, Figure? Similar in lines 297, 299, 304. Figure 5 should be mentioned in the start or middle of line 293, not in 307, need to revise.
11. Section 3.1.1 and section 3.1.2, 3.2 present very general data representation without major findings or limitations.
12. The discussion sections are also not showing in depth information. For example, in lines 388-390, In our study, 60 soybean plant images from different perspectives were collected at four different heights during image acquisition, so the 3D model obtained after 3D reconstruction was more accurate. How author confirm it? Also there was no mention about four height in the image acquisition section. Same in line 391-394, it should be very common phenomena. Also the amount of generated point vary with many other reason (such as: view angle, overlap (%), image quality, etc. How author define the density of points?
13. The datasets were prepared in very high controlled conditionunder artificial lighting conditions. How the data can be evaluated under natural condition or in the field condition. Author should describe this issue in the discussion. Does this data set compatible with other AI point cloud algorithms or only compatible with RandLA-Net and 434 BAAF-Net. Need to explain the issue.
14. Revise the conclusion explaining the methods, findings and limitation of the proposed datasets preparation with future scopes.
15. All figures need to be improved with better resolution/quality.
The English is moderate. Need to check grammar, and moderate editing of the English language is required. Also, check spelling, spacing, commas, and other formatting.
Reviewer 2 Report
1. Introduction part does not provide sufficient background information for the rationale and purpose of this study. Please expand and elaborate significantly on why this study has been conducted for readers.
2. Introduction provides a significant amount of previous studies; however, it is rather exhaustive and takes up a considerable portion of the manuscript. Please revise the literature review to be more concise and focus more on the purpose of this study.
3. Manuscript requires minor grammatical corrections, especially on noun strings.
4. What was the new finding or novelty of this study? It is difficult to comprehend, and the conclusion part only describes the results. It has to be expanded along with the rationale of the study.
5. Other minor comments are as follows:
L30 “analysis technology” - “technology” seems unnecessary
L30 “a challenging hot …” - seems repetitive.
L31 “is the basis for studying …” - It is hard to agree with this expression. please revise the sentence.
L32-33 “the very few … are difficult to obtain” - please revise the sentence (double negative).
L33-34 “collective organ instance segmentation benchmark data” - please revise this noun string to clarify the meaning of the sentence.
L36 “manpower” - please revise the word. i.e., “manual processing,” etc.
L37 “require” for which purpose? please specify the sentence.
L40 “is a key step toward …” - It is not explained why it is a key step.
L42 “well-labeled” - (1) is it different from the expression “well marked” in L36,59? (2) why and how labeling the point cloud facilitates the segmentation of point cloud data? Please explain it beforehand to readers
L44 - revise noun strings.
L45 “gave each point a real label” - (1) gave each “segment” a label? (2) what is the “real” label?
L46 “used… to supervise the machine learning algorithm” - supervised which task of ML? please specify.
L46-47 “and the real ground data to evaluate the …” - please revise the sentence. Seems to be a grammatical error (regarding the subject of the previous sentence).
L49 “organ level point cloud automatic segmentation” - noun string
L77-82 does it refer to the “photogrammetry” method? It would be better to introduce the types of three-dimensional reconstruction methods first.
L85 “low sunlight conditions in the laboratory” - meaning is somewhat confusing. low light conditions in the lab without sunlight? or low sunlight in indoor labs? please elaborate.
L87-92 seems repetitive. please revise it to be more concise.
L95 “solving” - considering the leaf density in a plant canopy, it can be enhanced, but I doubt it can be “solved.”
L120 “multiple view stereo method (MVS)” - would it be better to describe it as “multi… (MVS) photogrammetric method?”
L122 “silhouette contour principle” - what is it? please explain it beforehand.
L124-125 please revise the sentence. i.e., “obtained 3D models of soybean were manually labeled using CloudCompare software.”
L128-129 Why were these done in this study? please expand and elaborate on the purpose of this study in the background/introduction part.
Manuscript requires minor grammatical corrections, especially on noun strings.
Reviewer 3 Report
The investigation of plant phenotypes through the utilization of three-dimensional models has emerged as a significant avenue of inquiry for the automated acquisition of plant phenotypes. The present investigation involves the selection of five distinct soybean varieties, from which a total of 102 three-dimensional models of soybean plants are derived. These models are obtained through the application of multiple-view stereo technology (MVS) to reconstruct the entire growth period of soybeans, comprising 13 developmental stages. The authors declare promising results. However, a comparation with other method from SoA need to be done.
It will be good to place Table 1 on one page, Figure 6 is not with good quality.
Figures 4, 8 and 9 need to be more in deep explained.
Round 2
Reviewer 1 Report
This research article focused on three dimensional annotated datasets for Soybean for 3D segmentation of plant organs. The author improved the manuscript well. I will request to consider the following items to update the quality.
Comments:
1. There is no need to make section in abstract like (background, methods, Results, conclusion etc. with numbering. Please revise.
2. Check the typos in the manuscripts. Like in L245, “was stored in. txt format.” should be “was stored in .txt format.”
3. “rgb” can be written as RGB.
4. Improve the resolution of Figure 5.
5. Figures and table representation can be improved.
Author need to focus on the quality of English writing specially the sentence makings and typos error.
